# Physical and Mechanical Properties of High-Strength Concrete Modified with Supplementary Cementitious Materials after Exposure to Elevated Temperature up to 1000 °C

**DOI:** 10.3390/ma13030532

**Published:** 2020-01-22

**Authors:** Jianwei Zhou, Dong Lu, Yuxuan Yang, Yue Gong, Xudong Ma, Baoying Yu, Baobao Yan

**Affiliations:** 1Building Materials Science Academy of China West Construction Group Co., Ltd, Chengdu 610213, China; yby2872@163.com; 2School of Materials Science and Engineering, Xi’an University of Architecture and Technology, Xi’an 710055, China; yyx9202@163.com (Y.Y.); 20120949@cqu.edu.cn (X.M.); 3School of Materials Science and Engineering, Chang’an University, Xi’an 710064, China; dongluhit@163.com (D.L.); yolandagongyue@163.com (Y.G.); bby1124@163.com (B.Y.); 4School of Civil Engineering, Harbin Institute of Technology, Harbin 150000, China

**Keywords:** high-strength concrete, supplementary cementitious materials, elevated temperature, physical properties, relation

## Abstract

This paper presents the experimental findings of a study on the influence of combining usage of supplementary cementitious materials (SCMs) on the performance of high-strength concrete (HSC) subjected to elevated temperatures. In this study, four types of HSC formulations were prepared: HSC made from cement and fly ash (FA), HSC made from cement and ultra-fine fly ash (UFFA), HSC made from cement and UFFA-metakaolin (MK), and HSC made from cement and FA-UFFA-MK. Mechanical and physical properties of HSC subjected to high temperatures (400, 600, 800, and 1000 °C) were studied. Furthermore, the relation between residual compressive strength and physical properties (loss mass, water absorption, and porosity) of HSC was developed. Results showed that the combined usage of SCMs had limited influence on the early-age strength of HSC, while the 28-d strength had been significantly affected. At 1000 °C, the residual compressive strength retained 18.7 MPa and 23.9 MPa for concretes containing 30% UFFA-5% MK and 10% FA-20% UFFA-5% MK, respectively. The specimen containing FA-UFFA-MK showed the best physical properties when the temperature raised above 600 °C. Combined usage of SCMs (10% FA-20% UFFA-5% MK) showed the lowest mass loss (9.2%), water absorption (10.9%) and porosity (28.6%) at 1000 °C. There was a strongly correlated relation between residual strength and physical properties of HSC exposed to elevated temperatures.

## 1. Introduction

These days, high-strength concrete (HSC) has been widely used for building construction, bridges, and dams due to its super strength, excellent durability, and competitive cost [1,2]. However, it presents weak high-temperature resistance considerably limiting its potential [3,4].

Concrete consists of cement, aggregate, water, and admixtures, if we focus the effects of raw materials on the performance of concrete, we can note that the pore structure and the volume fraction of aggregate have a significant effect on properties of concrete [5,6]. In high-temperature conditions, concrete is not stable due to the decomposition of aggregate and deformation of internal physical/chemical changes, leading to degrade and damage of concrete [2,6,7]. Previous studies revealed that the microstructure of cement-based materials changes with temperature [8,9,10,11], the disappearance of ettringite (AFt) at 100 °C, a decomposition of CH and C-S-H at 400–600 °C, and a transformation of C-S-H into the nesosilicate form could be found beyond 750 °C [7]. Hydration products originally from cementitious materials are mainly responsible for the strength of concrete, it can fill pores and delay concrete cracking. Therefore, from the perspective of raw materials, optimizing the cementitious materials could be more effective than aggregate gradation in increasing the heat resistance of HSC.

Fly ash (FA) is one of the most commonly used supplementary cementitious materials (SCMs), the effect of FA on the performance of HSC subjected to elevated temperatures has been attracted the interest of researchers since the 1960s [12,13]. Wanget al. [14] carried out an experiment to investigate the effect of FA on mechanical properties and thermal conductivity of HSC, and found that compressive strength reduced about 26% at 550 °C, and the thermal conductivity increased 22% when the relative humidity was 100%. Valencia et al. [12] indicated that the residual strength of the FA/GBFS and FA/OPC are 15 and 5.5 MPa, respectively at 1100 °C. Aydinet al. [13] showed that the superior properties of FA specimen may be attributed to the strong aggregate–cement paste interfacial transition zone (ITZ) at 900 °C. Furthermore, a series of studies were conducted to investigate the effect of fuel ash [4], high volume fly ash [3,15] and volcanic ash [16] on the performance of HSC exposure to high temperatures. Furthermore, metakaolin (MK) as a kind of high activity SCMs, which can form anhydrous aluminum silicate (Al_2_O_3_·2SiO_2_) by dehydration at temperatures between 600 °C and 900 °C, making it possible to improving properties of HSC [15,17,18,19,20,21]. Oswaldoet al. [17] comparative properties of alkali activated MK cement pastes exposed to high temperatures, and indicated that the specimen remained more than 30 MPa at 800 °C. Nikhil et al. [18] investigated the effect of FA and MK on the performance of concrete, and revealed that a 2% addition of MK decreased porosity by 10%. Apart from mentioned above, some reports indicated that the addition of ground granulated blast furnace slag (GGBS) [22], ground granulated blast furnace furnace (GGBF) [23,24], alkali activated slag [3,17,25], and biochar [8,26] can all increase the properties of HSC to a certain extent. Although a large number of inorganic admixtures have been investigated as SCMs for improving the properties of concrete, there is a limited exploration into the role of combining of FA-UFFA-MK, in modifying mechanical and physical properties of HSC subjected to elevated temperatures. Therefore, to investigate the synergistic effect of adding a blend of FA, UFFA, and MK on the properties of HSC subjected to elevated temperatures is needed indeed.

This paper, for the first, provides a comprehensive understanding of mechanical characteristics, physical properties and the relation between mechanical performance and physical properties of HSC modified with SCMs (i.e., FA, UFFA, combined usage of UFFA and MK, and combined usage of FA, UFFA, and MK) subjected to elevated temperatures. Firstly, mechanical performance tests were carried out to measure compressive and flexural strength at the ambient temperature (AT), and residual and relative compressive strength at the elevated temperatures. Subsequently, physical tests were conducted to estimate the influence of combined usage of SCMs on the physical performance of HSC subjected to elevated temperatures, including physical appearance, loss of mass, water absorption, and porosity. Finally, the relation between residual compressive strength and physical properties of HSC was developed. The effects of combining usage of SCMs on the mechanical and physical properties of HSC were deeply interpreted in this study. It is exceedingly significant to promote HSC to be widely used in construction and building engineering.

## 2. Experimental Details

### 2.1. Materials

#### 2.1.1. Cement and SCMs Used

Ordinary Portland cement (P·O 52.5) confirmed to GB 175–2007 used for all mixtures [27]. Table 1 presents the properties of cement.

Figure 1 and Figure 2 show the scanning electron micrograph and particle size distribution of SCMs. Figure 1a,b present the spherical shape of the FA particles. These particles act as a lubricant and improve the slump of the mixture. Table 2 shows the chemical composition of cement and SCMs, which complied with the requirements of the Chinese Standards [28,29,30].

#### 2.1.2. Aggregate

Crushed quartz stone with a continuous grading used as coarse aggregate. It has a crushed index of 5.2%, a specific gravity of 2.6 g·cm^−3^, a absorption of 0.3% and a maximum size of 10 mm, respectively. River sand with a fineness modulus of 2.6 (medium sand) used as fine aggregate. It has a water absorption of 1.2% and a specific gravity of 2.5 g·cm^−3^, respectively. The aggregate complied with the requirements of the Chinese Standards GB/T 14684–2011 and GB/T14685–2011 [31,32].

#### 2.1.3. Admixture

Polycarboxylate-based superplasticizer (SP) with 35% solid content and a water-reducing ratio of 28%, which used as an admixture in 1% proportions of cementitious materials weight.

### 2.2. Mixture and Preparation

Five different mix proportions were considered in this paper. Noted that, all mix design parameters were maintained constant besides the SCMs. Table 3 and Table 4 present the mix proportions of the control mix and HSC, respectively.

### 2.3. Test Methods

The slump value was used to measure the workability of fresh mixture, according to the Chinese Standards GB/T 50080–2016 [33]. 

The mechanical properties of concrete at the AT according to the Chinese Standards GB/T 50081–2002 [34]. For compressive and flexural strength test, a 100 mm × 100 mm × 100 mm cubic mold and a 100 mm × 100 mm × 400 mm prism mold were used, respectively. Specimens were demolded after 24 h of casting and kept in a curing room (20 ± 1 °C, RH ≥ 95%) before testing. All concretes were tested after 3, 7, and 28 d of curing, respectively.

The mechanical performance of concrete subjected to elevated temperatures was characterized by residual compressive strength [1,4,26]. For residual compressive strength test, a 100 × 100 × 100 mm cubic mold was used, the specimens were dried to constant weight after 28 d of curing. The concretes were heated in an electrically controlled furnace (Shanghai Meiyu Instrument Equipment Co., Ltd., Shanghai, China) to the setting temperatures (400, 600, 800, and 1000 °C), the heating rate was 3 °C min^−1^ and then kept at a constant temperature for 3 h. Finally, the furnace was turned off and the samples were left until cooled to the AT (as seen in Figure 3) [4].

The physical properties of the specimen subjected to elevated temperatures were characterized by physical appearance, water absorption by immersion, loss mass, and porosity [1,26]. All concretes were tested after 28 d of curing, according to the Chinese Standards YB/T 5200–1993 [35].

## 3. Results and Discussion

### 3.1. Workability

Figure 4 shows the slump value of fresh concrete. As expected, the addition of SCMs resulted in an increment of slump value, compared with the control mix, especially for the S1-HSC (addition of FA). This could be ascribed to the spherical shape of the FA particles [23]. Which act as a lubricant and improve the slump of mixture [22]. As illustrated in Section 2.1.1.

The results can also be seen that the HSC modified with FA (S1-HSC) showed better workability than that of the S2-HSC. Similarly, the slump value of S1-S2-S3-HSC containing 10% FA was slightly higher than that of S2-S3-HSC. Indicating that the FA could be more effective than UFFA and MK in increasing the workability of HSC. This mainly attributed to the shape and size of FA particles result in reducing obstruction of the mixture [22,23].

### 3.2. Mechanical Properties at the AT

#### 3.2.1. Compressive Strength

Figure 5a shows the compressive strength of concrete. As expected, the compressive strength of HSC modified with SCMs was higher than that of the control mix. Specimens increased by 0.3% (S1-HSC), 0.9% (S2-HSC), 3.8% (S2-S3-HSC), and 4.3% (S1-S2-S3-HSC) in each case, compared to the control mix after 3 d of curing. While increased by 12.6% (S1-HSC), 14.2% (S2-HSC), 19.0% (S2-S3-HSC), and 22.2% (S1-S2-S3-HSC), respectively, after 28 d of curing. Indicating that the influence of SCMs on early-age compressive strength was limited, while it had a significant influence on long-term age strength. This could be attributed to the lower reactive of SCMs during early-age hydration, the hydration products increase with the curing time, a large number of hydration products formed after 28 d of curing.

It can also be obtained that the compressive strength of S1-HSC and S2-HSC both approximately increased by 14%, while combined usage of SCMs modified HSC (S2-S3-HSC and S1-S2-S3-HSC) approximately increased by 20%, compared with the control mix after 28 d of curing. Indicating that the combined usage of SCMs further exacerbated an increment in the compressive strength. This could be attributed to the possesses pozzolanic properties of SCMs [18]. Furthermore, the micro-filling capacity (as seen in Figure 2) of SCMs occupies the voids in the concrete [15].

#### 3.2.2. Flexural Strength

The effect of SCMs on flexural strength at AT was similar to that of the compressive strength. The HSC modified with SCMs showed higher flexural strength than that of the control mix. As shown in Figure 5b. Flexural strength increased by 0.9% (S1-HSC), 1.8% (S2-HSC), 3.6% (S2-S3-HSC), and 5.9% (S1-S2-S3-HSC), respectively, after 3 d of curing, while increased by 9.0% (S1-HSC), 11.3% (S2-HSC), 12.3% (S2-S3-HSC), and 19.2% (S1-S2-S3-HSC), respectively at 28 d.

### 3.3. Mechanical Properties Subjected to Elevated Temperatures

From Figure 6, it can be seen that two distinct stages of evolution of residual compressive strength of the specimen subjected to elevated temperatures. Residual compressive strength of HSC first increased and then decreased with the increased of temperature. The first stage (AT-400 °C) was characterized by a slight rise in residual compressive strength. Compared with the specimens at the AT. After heating to 400 °C, specimens increased by 0.3% (control mix), 4.5% (S1-HSC), 5.2% (S2-HSC), 5.9% (S2-S3-HSC), and 10.4% (S1-S2-S3-HSC), respectively. This could be attributed to the fact that the water result in the decomposition of hydration products lets to rehydrate the anhydrous cement, increases the amount of C-S-H gel and improves the strength of the concrete at 400 °C [12,15,18,20,21,36].

The second stage (400–1000 °C) was characterized by a loss of residual compressive strength of concrete. It can be observed that the residual compressive strength ratio of HSC was about 35–70% at 800 °C, while the temperature reached 1000 °C, the residual compressive strength ratio was 9.9% for the control mix, 11.5% for S1-HSC, 14.3% for S2-HSC, 18.7% for S2-S3-HSC, and 23.3% for S1-S2-S3-HSC, respectively. This could be attributed to the deterioration of paste under high temperature [1]. Also, residual compressive strength retained 18.7 MPa and 23.9 MPa for concretes containing 30% UFFA-5% MK and 10% FA-20% UFFA-5% MK, respectively, after heating to 1000 °C. Indicating that the combined usage of SCMs had a positive effect on the residual strength of HSC at 1000 °C for 3 h, due to the super pozzolanic properties and micro-filling capacity of combining usage of SCMs [21,37]. This was supported by the scanning electron micrograph and particle size distribution of SCMs (as Figure 1 and Figure 2).

### 3.4. Physical Properties Subjected to Elevated Temperatures

#### 3.4.1. Physical Appearance

Figure 7 showed the appearance of the samples subjected to elevated temperatures. Up to 400 °C, the concretes maintained the color (gray-black) of the samples and their structural integrity. Concretes began to damage and crack at 600 °C due to the decomposition of C-S-H gel and CH [1,20]. Up to 800 °C, the color of the samples changed to whitish grey, and the specimens had visible cracks. When the temperature increased to 1000 °C, the control mix and the HSC modified with FA almost completely destroyed, while the HSC containing FA-UFFA-MK showed a relatively complete structure.

#### 3.4.2. Loss Mass

Three distinctive stages of evolution of loss mass of the concrete with temperature can be obtained, as seen in Figure 8. The first stage was corresponding to the temperature range from AT to 400 °C, and was characterized by a sharp increase of loss mass of the specimens. In this stage, the loss mass of concrete at 400 °C approximately increased by 10 times than at the AT. This mainly due to the moisture loss [26], including the departure of the free water and the combined water, and the decomposition of C-S-H gel and aluminate hydrates [10]. The second stage was corresponding to the temperature range from 400 °C to 600 °C. In this stage, the loss mass stabilized with temperature. The third stage was characterized by a significantly increased in loss mass beyond 600 °C. This can be explained by the fact that the concrete began to damage and crack at 600 °C due to the destruction of C-S-H, decomposition of CH and desiccation of pore structure [13].

At 400 °C, compared with the control mix, the loss mass of specimens slightly increased by 0.3% (S1-HSC), 1.3% (S2-HSC), 1.7% (S2-S3-HSC), and 1.9% (S1-S2-S3-HSC), respectively. This could be ascribed to the water from the decomposition of some hydration products at 400 °C, resulting in rehydrate the anhydrous cement particles [1]. Besides, the highest loss mass observed on S1-S2-S3-HSC, this can be explained by the highest content in fineness and C-S-H gel. At 600 °C, the HSC modified with SCMs showed lower loss mass than the control mix due to the pozzolanic action involves the consumption of CH [26]. At 1000 °C, loss mass of the specimens decreased by 3.7% (S1-HSC), 8.4% (S2-HSC), 17.0% (S2-S3-HSC), and 25.3% (S1-S2-S3-HSC), respectively, compared with the control mix due to the addition of SCMs can provide an excellent effects on loss mass, especially, the combined usage of FA-UFFA-MK can further restrain the loss mass of concrete beyond 600 °C. This was also supported by strength results, as illustrated in Section 3.3.

#### 3.4.3. Water Absorption

Change in water absorption subjected to elevated temperatures in concrete can be distinguished into two distinct stages of evolution: a slight increment in the water absorption between AT and 600 °C, and a significant increment in the water absorption beyond 600 °C, as seen in Figure 9. This could be attributed to the surface of concrete cracked and the internal structure of specimen damaged up to 600 °C, and the cracking and damage further intensified after heating to 800 °C (as seen in Figure 7). Furthermore, the damaged of the concrete resulted in an increment of water absorption related to reducing the strength of the specimen, this will be illustrated in Section 3.5. This was also supported by the observation of microstructure through SEM imaging. The HSC modified with SCMs showed a denser microstructure than that of the control mix.

As expected, the water absorption of HSC modified with SCMs was lower than that of the control mix. The control mix and concrete with 10% FA (S1-HSC) showed a steep rise in water absorption at 600 °C, while concrete containing UFFA and combined usage of SCMs (S2-S3-HSC and S1-S2-S3-HSC) showed relatively lower water absorption compared to the control concrete and HSC-S1. The water absorption of specimens decreased by 7.3% (S1-HSC), 16.1% (S2-HSC), 18.8% (S2-S3-HSC), and 23.2% (S1-S2-S3-HSC) in each case, compared with the control mix at 1000 °C. This could be attributed to the micro-filling effect and pozzolanic reaction of SCMs. Indicating that the combined usage of SCMs can provide a significant effect on the water absorption of the specimen subjected to elevated temperatures.

#### 3.4.4. Porosity

The porosity of the specimen showed an increasing trend as the temperatures, as seen in Figure 10. Poonet al. [20] and Daset al. [8] showed that the concrete exposed to high temperature results in pore coarsening, resulting in drastic loss strength and high permeability. At 400 °C, the control mix showed a steep rise in porosity, while the HSC modified with SCMs showed relatively lower porosity. When the temperature heated to 600 °C, the porosity of HSC reached about 60% of the specimen at 1000 °C. This can be attributed to the differential dilatation between aggregate and cement paste, and decomposition of the hydration products creates an additional porosity by micro-cracks generate. Similar results have been proved by Gutpa et al. [26].

The HSC modified with SCMs showed a lower porosity than that of the control mix at the same temperature due to the addition of SCMs could restrain the destruction of hydrates to some extent (Figure 11). Compare the porosity of HSC subjected to elevated temperatures, the S1-HSC presented the highest porosity, following by the S2-HSC, S2-S3-HSC, and S1-S2-S3-HSC. Indicating that the combined usage of SCMs provided a positive effect on reducing the porosity of concrete.

### 3.5. Relation Between Residual Compressive Strength and Physical Properties

Figure 12 presents the relation between residual compressive strength and physical properties (mass loss, water absorption, and porosity) of concrete subjected to elevated temperatures.

#### 3.5.1. Relation Between Residual Compressive Strength and Loss Mass

A slight increment in the residual compressive strength of the control mix for a loss mass at 5.4% and of the HSC for a loss mass about 5.5%, as seen in Figure 12a. This could be attributed to the higher weight of free water volatile due to the addition of SCMs at 400 °C. A sharp drop of residual compressive strength could be obtained for a loss mass at 6.9% (control mix), 5.6% (S1-HSC), 5.4% (S2-HSC), 5.5% (S2-S3-HSC), and 5.3% (S1-S2-S3-HSC), respectively. At 1000 °C, the mass loss retained 12.3% (control mix), 11.9% (S1-HSC), 11.3% (S2-HSC), 10.3% (S2-S3-HSC), and 9.2% (S1-S2-S3-HSC), respectively. Indicating that the combined usage of SCMs can provide excellent synergetic effects on loss mass of mixtures exposed to high temperatures.

#### 3.5.2. Relation Between Residual Compressive Strength and Water Absorption

A slight improvement in the residual compressive strength of the control mix for a water absorption at 3.0% and of the HSC modified with SCMs for a water absorption at 1.9–2.7%, as presented in Figure 12b. A sharp drop in the residual strength of the control mix for a water absorption could be found at 6.2% and of the HSC for a water absorption at 3.5–5.9%, due to the addition of SCMs with large specific surface area increases the water absorption of concrete at 400 °C. While the control mix indicated a higher water absorption than that of the HSC modified with SCMs at 600 °C, this could be attributed that the internal structure and the surface of specimen begin to form micro-cracks (as discussed in Section 3.4.1).

#### 3.5.3. Relation Between Residual Compressive Strength and Porosity

A slight increment in the residual strength in the specimen for a porosity about 15% (the control mix) and at 12% (HSC), respectively. These pores release the built-up of internal pressure during the subject at high temperatures, which decreased the damage of internal structure in concrete. For the HSC modified with SCMs, a sharp drop residual compressive strength of the specimens for a porosity at 19.0% (S1-HSC), 18.3% (S2-HSC), 17.8% (S2-S3-HSC), and 15.2% (S1-S2-S3-HSC), respectively. Indicating that the less porous the concrete, the higher its resistance. The augmentation of porosity generates a crack in the concrete and increases with temperature due to the decomposition of some hydration products.

## 4. Conclusions

An experimental study was conducted to investigate the effect of combined usage of SCMs on the workability, and mechanical and physical properties of concrete, developing the relationship between residual compressive strength and physical properties of concrete subjected to elevated temperatures. The following conclusions can be made:(1)The slump value of fresh concrete increased with addition of SCMs, compared with the control mix. The addition of FA showed more effective than that of the UFFA or combined usage of SCMs in increasing the workability of HSC.(2)The influence of SCMs on early-age (3 d) compressive strength of HSC was limited, while it had a significant influence on the long-term strength of HSC. The combined usage of SCMs can provide excellent synergetic effects on mechanical properties of HSC.(3)Two distinct stages of evolution of residual strength of the specimen subjected to elevated temperatures: a slight increase of residual compressive strength at 400 °C, and a sharp drop beyond 400 °C. The combining usage of SCMs had a positive effect on the residual compressive strength of HSC, for specimens containing UFFA-MK and FA-UFFA-MK at 1000 °C, the residual strength retained 18.5 MPa and 23.3 MPa, respectively.(4)The loss mass of concrete at 400 °C increased approximately by 10 times that of at the AT. The specimen containing FA-UFFA-MK showed the best physical properties when the temperature rose above 600 °C. A strongly correlated relation between residual strength and physical properties of HSC exposed to elevated temperatures.

## Figures and Tables

**Figure 1 materials-13-00532-f001:**
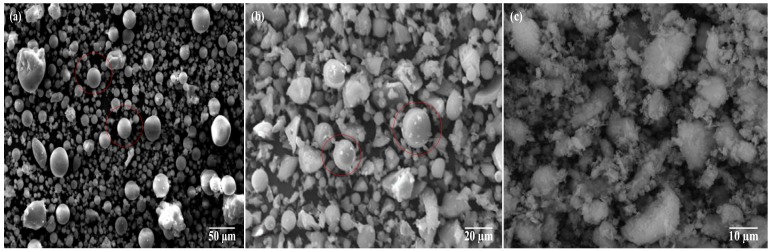
Scanning electron micrograph of SCMs: (**a**) FA, (**b**) UFFA, and (**c**) MK.

**Figure 2 materials-13-00532-f002:**
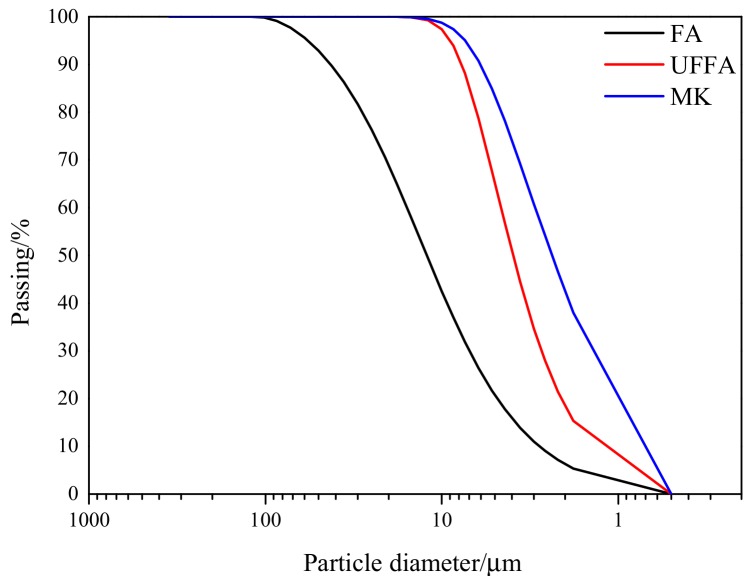
Particle size distribution of SCMs in this study.

**Figure 3 materials-13-00532-f003:**
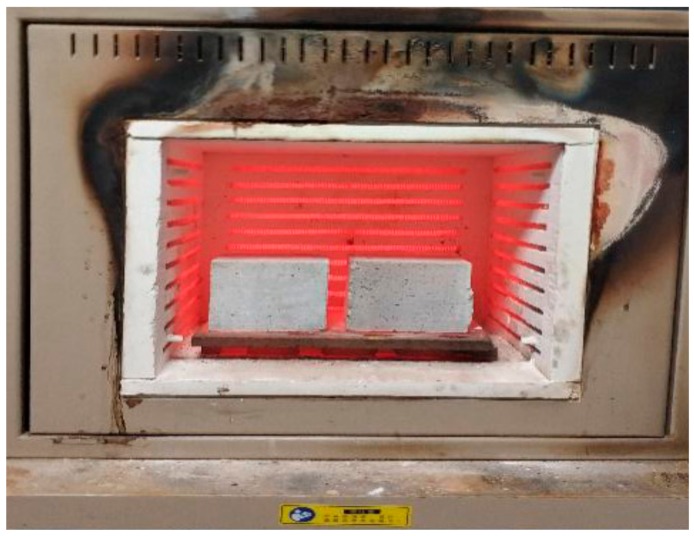
Specimens in an electrically controlled furnace.

**Figure 4 materials-13-00532-f004:**
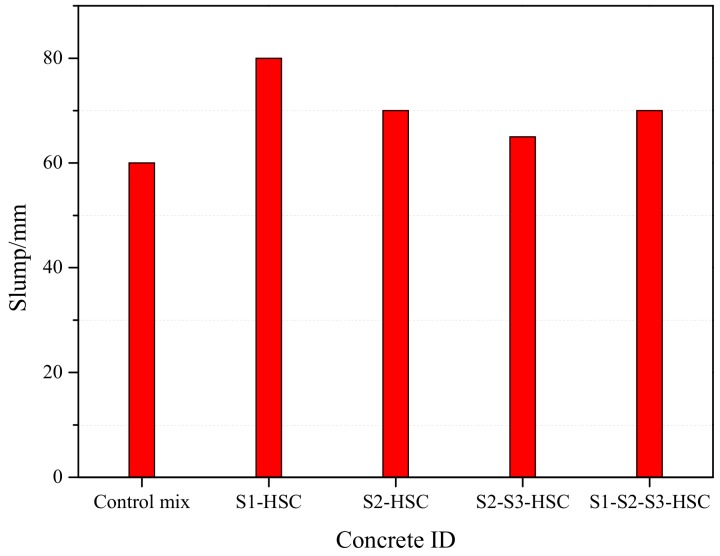
Slump of the mixtures.

**Figure 5 materials-13-00532-f005:**
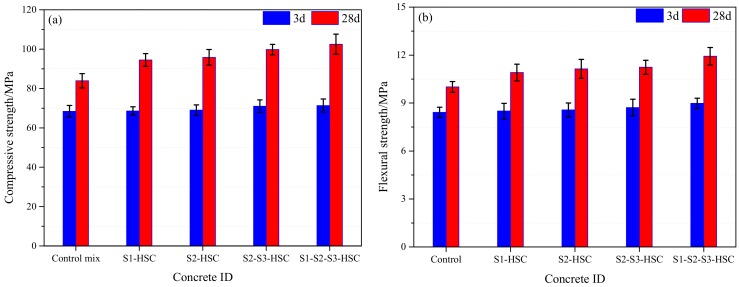
Mechanical properties of concrete at the AT: (**a**) compressive and (**b**) flexural strength.

**Figure 6 materials-13-00532-f006:**
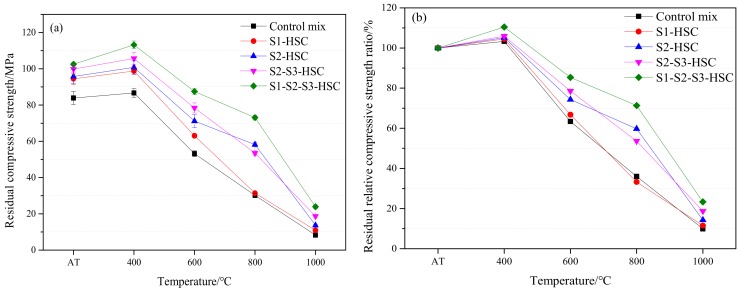
Mechanical properties of concrete subjected to elevated temperatures: (**a**) residual compressive strength and (**b**) residual relative compressive strength ratio.

**Figure 7 materials-13-00532-f007:**
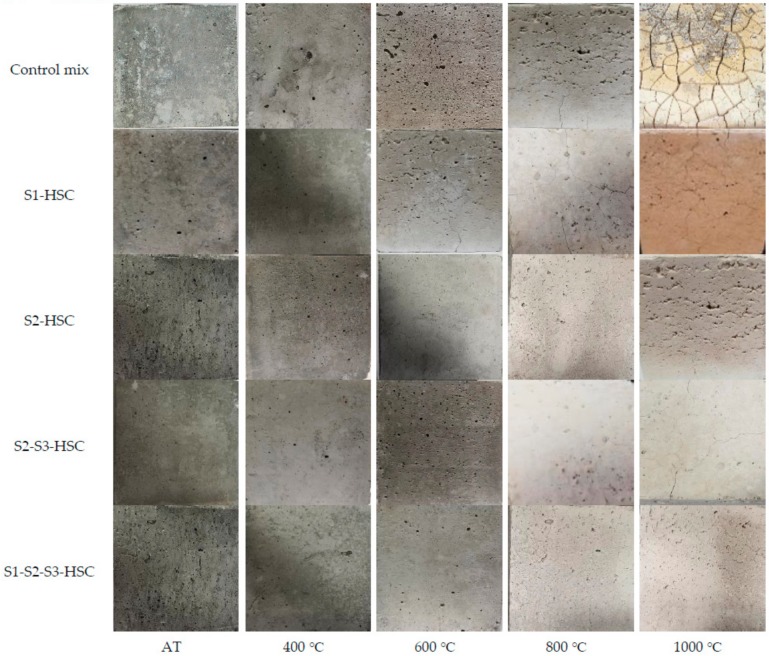
Appearances of specimen subjected to elevated temperatures.

**Figure 8 materials-13-00532-f008:**
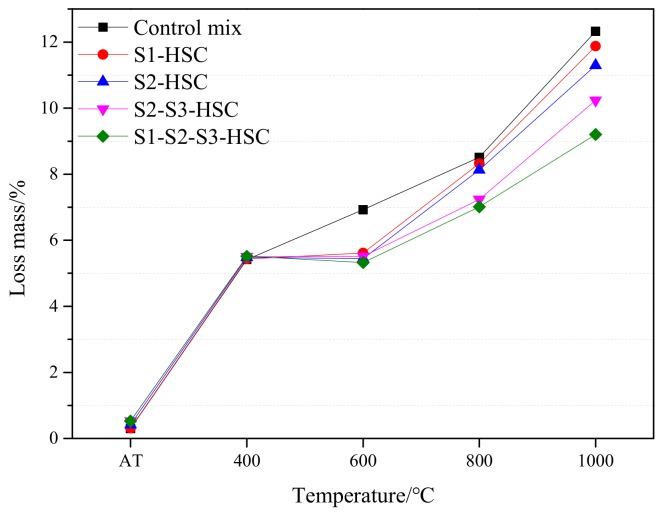
Loss mass of the specimens subjected to elevated temperatures.

**Figure 9 materials-13-00532-f009:**
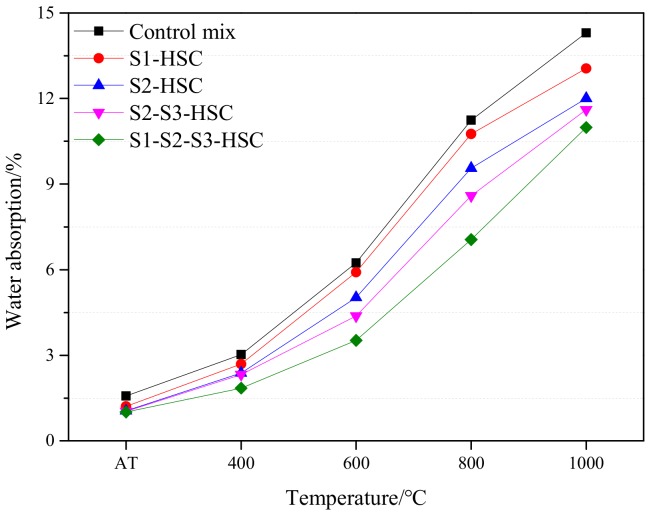
Water absorption of the specimens subjected to elevated temperatures.

**Figure 10 materials-13-00532-f010:**
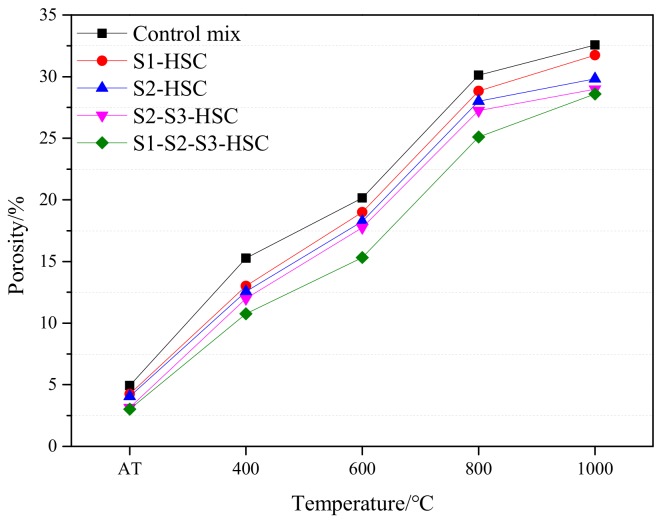
Porosity of the concrete subjected to elevated temperatures.

**Figure 11 materials-13-00532-f011:**
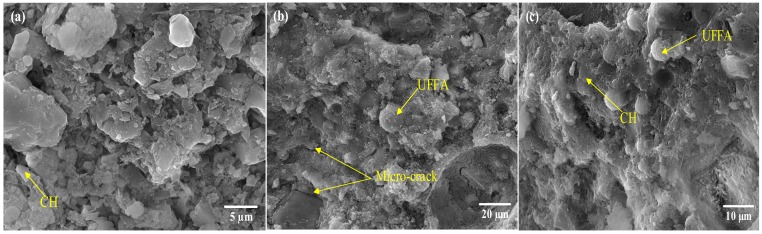
Microstructure of the (**a**) control mix, (**b**) 10% UFFA, and (**c**) 5% MK under 400 °C.

**Figure 12 materials-13-00532-f012:**
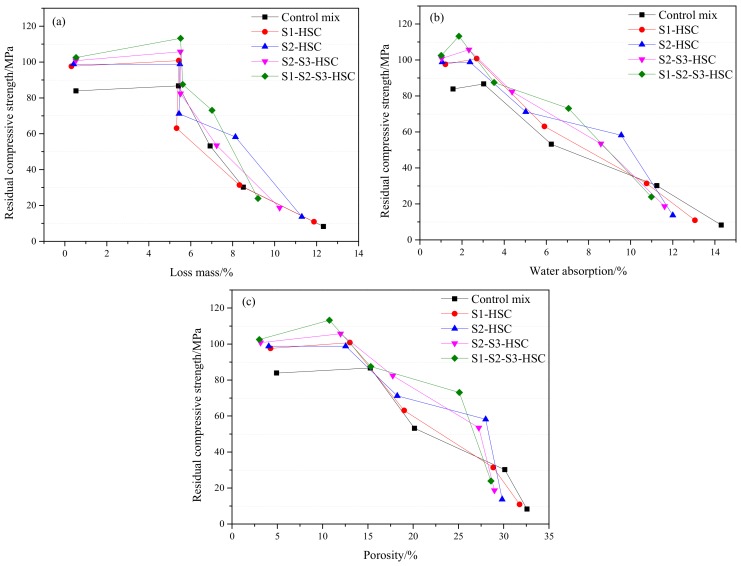
Relation between residual compressive strength and physical properties: the relation between residual strength and (**a**) loss mass, (**b**) water absorption, and (**c**) porosity.

**Table 1 materials-13-00532-t001:** Properties of cement.

Physical Properties		Setting Time /min		Flexural Strength /MPa		Compressive Strength /MPa	
Density /(g·cm^−3^)	Specific Surface /(cm^2^·g^−1^)	Initial	Final	3 d	28 d	3 d	28 d
3.05	4610	113.0	146.0	5.7	9.2	28.7	60.1

**Table 2 materials-13-00532-t002:** Chemical composition of cement and SCMs (mass%).

Label	CaO	SiO_2_	Al_2_O_3_	Fe_2_O_3_	SO_3_
Cement	59.7	21.5	5.8	4.0	2.0
FA	4.1	50.8	25.0	2.6	0.3
UFFA	4.4	59.5	28.6	2.6	0.3
MK	0.4	46.9	44.3	4.4	0.2

**Table 3 materials-13-00532-t003:** Mixture proportion of control concrete/(kg·m^−3^)

Cement	Water	Fine Aggregate	Coarse Aggregate	SP
600	126	634	1126	6

**Table 4 materials-13-00532-t004:** Concrete mixes with SCMs

Concrete ID	Mass of Cement Substituted by Cementitious Materials SCMs/%		
	FA	UFFA	MK
Control mix	0	0	0
S1-HSC	30	0	0
S2-HSC	0	30	0
S2-S3-HSC	0	30	5
S1-S2-S3-HSC	10	20	5

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
