# Peer review of "Physical and Mechanical Properties of High-Strength Concrete Modified with Supplementary Cementitious Materials after Exposure to Elevated Temperature up to 1000 °C"

_materials, 2020, doi:10.3390/ma13030532_

Round 1

Reviewer 1 Report

Page 3: from Table2 the oxide comoposition of cement is missing. Please give the oxide composion of cement too.

Page 8: Please explain the different porositys and colours on the Figure 7 I think in case of control mix above 500 °C it should be a lot of cracks (decomposition of Ca(OH)2 and CSH)

Page 9: Please explain the changing of  water absorption

Page 10 How you have measured the porosity? What is the influence the cracks?

Author Response

Dear editors and reviewers,

We are so deeply appreciate the time and effort you have spent in our manuscript. Thank you very much for your comments and suggestions. All your suggestions are very important, they have important guiding significance for my research. 

Best regards and happy new year.

Jianwei Zhou

Reviewer 2 Report

This article presents experimental study of the high-temperature effect on properties of concrete modified with supplementary  cementitious materials.

This paper deals with a topic of considerable practical importance to structural designers (for tunnels in particular). However, I have few, and I think irreparable criticism of this paper: The heating method used is not representative of real fire conditions, is not well monitored or controlled. It is not only the peak exposure temperature is important for concrete spalling, but rather the rate of heating and the thermal gradients created in the concrete. What is the cooling regime used? It is not described.

In addition, in my opinion, the definition of HPC in this paper is improper. Some experiment details are also not clear. Such as how many specimens were prepared and tested? These issues are important to the validities of the conclusions. At last, the writing of the whole paper should be further improved. For example, the conclusion section should avoid using the numbered items only. It should contain real conclusions and the plans for the future research — not the results summary only. The English also needs proof reading.

The title does not seem to be adequately defined. Please consider also changing " High Performance Concrete (HPC)" to "High Strength Concrete (HSC)". Table 1. Properties of cement - should be improved Table 4. Concrete mixes with SCMs - „Mass of SCMs substituted by cementitious materials/%” is not the right term, I think it should be:

„Mass of cement substituted by cementitious materials SCMs /%”

Abbreviations that have not been defined before are often used – for example AT; Fig 8 - wrong interpretation of loss mass results Incorrect chapter titles: „Mechanical Properties Subjected to Elevated Temperatures” „Physical properties Subjected to Elevated Temperatures”

Mistakes for example:

the concretes maintained the colour of the samples and thier tructural integrity……… (195)

shoud be?

the concretes maintained the colour of the samples and their structural

In conclusion, despite the fact the paper presents an interesting experimental campaign, it needs further work and comparison with existing literature to avoid misunderstandings.

Author Response

Dear reviewer,

we are so appreciate the time and effort you have spent in reviewing our manuscript. thank you for your comments and suggestions,all you comments and suggestions are very important,they have important guiding significance fou my research work.

Besr regards and happy new year! 

Jianwei Zhou

Round 2

Reviewer 1 Report

All changes were prepared in the articel.